# PathoRM: Computational inference of pathogenic RNA methylation sites by incorporating multi-view features

Hui Liu, Jiani Ma*, Xianjun Ma, Lin Zhang*

School of Information and Control Engineering, China University of Mining and Technology, Xuzhou, China

* jiani.ma@cumt.edu.cn (JM); lin.zhang@cumt.edu.cn (LZ)

## Abstract

Identifying pathogenic RNA methylation sites with a reasonable biological explanation has important implications for the treatment of diseases. Due to the limitations of in vitro experiments in identifying pathogenic RNA methylation sites, there is a growing need for computational workflows to enable accurate inference. Here, motivated by this profound meaning, we developed PathoRM, a biologically informed deep learning model, to infer associations between RNA methylation sites and diseases. PathoRM could provide convincing pathogenic RNA methylation sites and unravel the enigma of pathology in the epi-transcriptomic layer. PathoRM fuses RNA methylation host sequences and pathogenic descriptions as inputs, and subsequently employs large language models, multi-view learning algorithm, graph neural networks, an adversarial training approach, and "guilty-by-association"-derived negative sampling approach. PathoRM distils the semantically enriched feature embeddings, leading to more accurate and robust prediction performance across the metrics and datasets. Notably, incorporated with attention mechanism, PathoRM bestows itself biological interpretability through illuminating the dark matters in the host sequences of RNA methylation sites. This work is expected to assist in the discovery of pathogenic RNA methylation sites and conserved motifs, contributing to the advancement of genome research. Codes and pre-trained model are accessible at https://github.com/jianiM/PathoRM.

## Author summary

RNA methylation (RM) is a pivotal epi-transcriptomic modification that alters RNA nucleotides through methyl group additions, profoundly impacting gene expression, cellular differentiation, and essential biological processes crucial for maintaining cellular function. Dysregulation of RM is intricately linked to various diseases. Given the challenges associated with identifying pathogenic

**Data availability statement:** m6ADA dataset and m7GDA dataset we used in this study are provided on the github at https://github.com/jianiM/PathoRM.

**Funding:** This work was supported by the National Natural Science Foundation of China (Research Projects Nos. 61971422 to LZ, 31871337 to HL). The funders had no role in study design, data collection and analysis, decision to publish, or preparation of the manuscript.

**Competing interests:** The authors have declared that no competing interests exist.

sites through laboratory examination, there is an urgent need for computational workflows capable of accurately inferring pathogenic RNA methylation sites to facilitate comprehensive biological investigations. Here, we developed PathoRM, a biologically informed deep learning model aimed at elucidating the associations among RNA methylation sites and diseases. PathoRM integrates RNA methylation host sequences and pathogenic descriptions using large language models, multi-view learning algorithms, graph neural networks, adversarial training, and a negative sampling method derived from "guilty-by-association" principles. By distilling semantically enriched feature embeddings, PathoRM achieves promising predictive accuracy and robustness across diverse metrics and datasets. Notably, even without explicit annotations for sites, PathoRM can capture the intrinsic pathogenic regions, which is overlapped with the conserved motif, in the RM host sequence, offering biological insights into the decision-making procedure.

## 1. Introduction

RNA methylation (RM) is a crucial epi-transcriptomic modification, fundamentally altering the chemical structure of RNA nucleotides by appending methyl groups [1]. The most common RNA methylation modifications include $m^6A$ ($N^6$-methyladenosine), $m^5C$ (5-methylcytidine), $m^7G$ ($N^7$-methylguanosine) and 2-O-methylation [2,3]. They exert profound effects on gene expression, cell differentiation, and essential biological processes which are important for maintaining cellular function and responding to environmental cues [4–6]. The dysregulation of RM is intricately associated with pathophysiological conditions such as cancer, neurological disorders, cardiovascular diseases, and metabolic disorders, emphasizing its critical role in both health and disease [7–9].

Given the major constraints in identifying pathogenic RM sites with conventional *in vivo* experiments, there arises a critical necessity for the development of computational workflows to accurately infer these sites, laying the groundwork for deeper biological investigations. Thanks to the advent of high-throughput sequencing technology, such as miCLIP [10], $m^6A$-CLIP-seq [11], and DART-seq [12], RM sites can be pinpointed with single-nucleotide resolution. This has led to the creation of comprehensive RM databases like RMVar [13], m7GHub [14], and RMDisease [15], serving as repositories for pathogenic RM sites and induced biological pathways. Leveraging them, great opportunity arises to design computational workflows capable of efficiently predicting potential pathogenic RM sites, thereby deepening our understanding of their role in disease pathogenesis.

Various works have indicated that machine learning methods could significantly advance this area. DRUM stands as the pioneering machine learning model for predicting the pathogenic RM sites [16]. It begins by building a multi-layer heterogeneous network containing "$m^6A$-gene-disease" associations and subsequently employs a random walk technique to infer pathogenic $m^6A$ sites. m7GDisAI [17], BRPCA [18], SpBLRSR [19] developed matrix decomposition and subspace learning models for prioritizing pathogenic $m^7G$ sites via predicting $m^7G$-disease associations.

Despite the promise of these approaches [17–19], some challenges remain for the inference of pathogenic RM sites, including (i) They may struggle with scalability when dealing with large-scale datasets. (ii) They lack flexibility with structured input format which may limit the flexibility on data structure, especially for disease descriptions where the underlying structure may not be well understand or difficult to encode manually. (iii) They totally rely on manual feature engineering or domain-specific knowledge, hindering their ability to capture complex patterns effectively. (iv) They highly rely on prior similarity information which is biased to some extent.

Confronted with the above-mentioned challenges, deep learning algorithms, esteemed for their remarkable data representation capabilities, have found widespread applications with successive layers of nonlinear transformations, adeptly handling diverse data structures like sequences, images, and irregular graphs. Harnessing deep-learning algorithms holds promise for further enhancing accuracy, scalability and flexibility for predicting and prioritizing pathogenic RM sites, and thereby unveiling cryptic patterns that are intricately linked to pathogenic RM sites. (iv) Additionally, for mitigating challenge multi-view learning could capture the complementary representations from comprehensive characterizations by integrating diverse features of RM sites or diseases, thus facilitating the acquisition of unbiased similarity information.

Although deep learning models or multi-view learning models haven't yet been specifically proposed for addressing the prioritization of pathogenic RM sites, numerous models have been developed and tailored for prioritizing potential targets, drugs, pathogenic miRNAs or lncRNAs [20–22]. They could offer valuable inspiration for the model design in inferring pathogenic RM sites. Typically, deep learning-based association prediction algorithms leverage convolutional neural network (CNN), recurrent neural network (RNN) and graph neural network (GNN) to extract complex features of miRNAs, drugs, lncRNAs, and targets. These features are then combined with multilayer perceptron (MLP), matrix factorization, or machine learning algorithms for final prediction. Notably, the selection of deep learning algorithm for molecular feature representation often depends on the data structure of association entities. For instance, DeepDTA [23] and DeepDCA [24] respectively utilize CNN and long short-term memory network (LSTM) to extract feature embeddings from drug Simplified Molecular Input Line Entry System (SMILES) sequences and protein primary sequences, subsequently employing MLP for inferring drug-target association prediction. HyperAttentionDTI [25] utilizes attention mechanism to dynamically adjust the feature embeddings of drug and protein sequences. In addition to sequence structures, drugs can also be represented by atomic interaction graphs, and proteins by their three-dimensional structures. GraphDTA [26] employs graph attention network (GAT), graph convolutional network (GCN), graph isomorphism network (GIN), and GAT-GCN algorithms to extract structural and topological features from drug atomic interaction graphs. DrugVQA [27] utilizes protein contact maps to represent amino acid residue distances in protein three-dimensional structures, employing dynamic CNN to capture spatial relationships between different residues within proteins, thus learning deep features of protein structures. Moreover, KGE_NFM [28] leverages knowledge graphs and recommendation systems for drug-target association prediction. It integrates heterogeneous information from multiple omics data sources via knowledge graph embedding, then integrates heterogeneous information through neural factorization machines, ultimately achieving accurate and robust predictions of drug-target associations.

Although deep learning models excel in handling drug or protein data structures, disease information in unstructured text descriptions is relatively complex and irregular, making direct feature extraction challenging. In recent years, the rapid development of large language models (LLMs) has offered new possibilities for directly extracting features from unstructured disease text information. Facilitated by various self-supervised tasks, models such as Transformers [29] or BERT [30] trained on vast biomedical corpora have acquired contextual understanding, enabling the extraction of rich embedding representations. Hence, LLMs are positioned to directly derive feature embeddings from textual disease information, thereby augmenting the effectiveness of deep learning models in addressing diverse prioritization tasks involving pathogenic molecules, including the prioritization of pathogenic RM sites.

Inspired by deep learning and multi-view learning algorithms, we designed and evaluated the performance of a LLM, multi-view learning and graph autoencoder-based workflow – called "PathoRM" – for inferring and prioritizing pathogenic RM sites. The main contributions of the paper are as follows:

(i) We employed a pre-trained DNA methylation site prediction LLM iDNA-ABF, biomedical pre-trained language model BioBERT [31], naïve multi-view learning algorithm [32], and a graph autoencoder, with adversarial training and negative sampling scheme, to predict pathogenic RM sites.

(ii) We utilized the attention mechanisms of iDNA-ABF to explore the link between motifs of RM host sequences and pathology.

(iii) This approach provides the prospect of being able to accurately infer the pathogenic RM sites, and offers a biological explanation for model's decision.

The remaining sections of this paper are organized as follows: Material and Methods section introduces data collection and processing procedure, workflow and methodology of PathoRM. Results section presents the experimental design, comparison experimental results, ablation studies and biological explanation. Discussion section summarizes the key findings, explains the implications as well as limitations, and proposes the future research based on PathoRM.

## 2. Materials and methods

### 2.1. Data collection and processing

Rooted in the biological premise that genetic variation at certain loci leads to abnormal RM expression and contributes to disease development,we curated two pathogenic RM site datasets: the m7G-Disease Association (m7GDA) dataset from m7GHub [14] and the m6A-Disease Association (m6ADA) dataset from RMVar [13] for downstream model evaluation. Taking m7GDA dataset as an example, the identification of pathogenic m7G sites is determined by whether disease-associated genetic variants result in the gain or loss of these sites. Leveraging 1218 variant sites as a bridge, we compiled 741 pathogenic m7G sites with high or medium confidence level, covering 177 diseases and 768 validated associations. Additionally, each pathogenic m7G site is represented by a pair of 41-bp sequences—one reference and one alternative—centered on the site, ensuring comprehensive site characterization. Similarly, m6ADA consists of 2860 associations among131 pathogenic m6A sites and 1338 disease. For each m6A site, a 65-bp reference sequence and a 65-bp alternative sequence centered on the site are provided.

### 2.2. The workflow of PathoRM

PathoRM is a biologically informed deep multi-view learning-based model. It is designed to predict pathogenic RM sites while also offering insights into the underlying pathogenesis through post-analysis of its decisions. Fig 1 showcases its computational workflow.

As for a RM-disease pair, PathoRM takes RM host sequence and disease text description as inputs, which is followed by well-trained LLM models iDNA-ABF and BioBERT, respectively. Subsequently, naïve multi-view learning integrates diverse RM features and disease features to generate the RM affinity matrix and disease affinity matrix in a metric-free and data-driven manner. Then taking RM sites and disease as nodes, the RM–disease heterogeneous network is initialized with known RM–disease associations, RM affinity matrix and disease affinity matrix. Finally, graph autoencoder is used to propagate sequence-level features among nucleotides, capture the low-dimensional feature embeddings, and prioritize the pathogenic RM sites according to the prediction scores. To achieve robust model, its computational pipeline is trained with adversarial training and "guilty-by-association"-derived negative sampling method. Here, we described the details of each module in the computational workflow as follows.

### 2.3. LLMs for RM sites and disease features extraction

iDNA-ABF is a multi-scale biological language model developed based on BERT for the prediction of DNA methylation sites [33]. It treats the DNA methylation host sequence as biological text and utilizes various k-mer strategies to segment

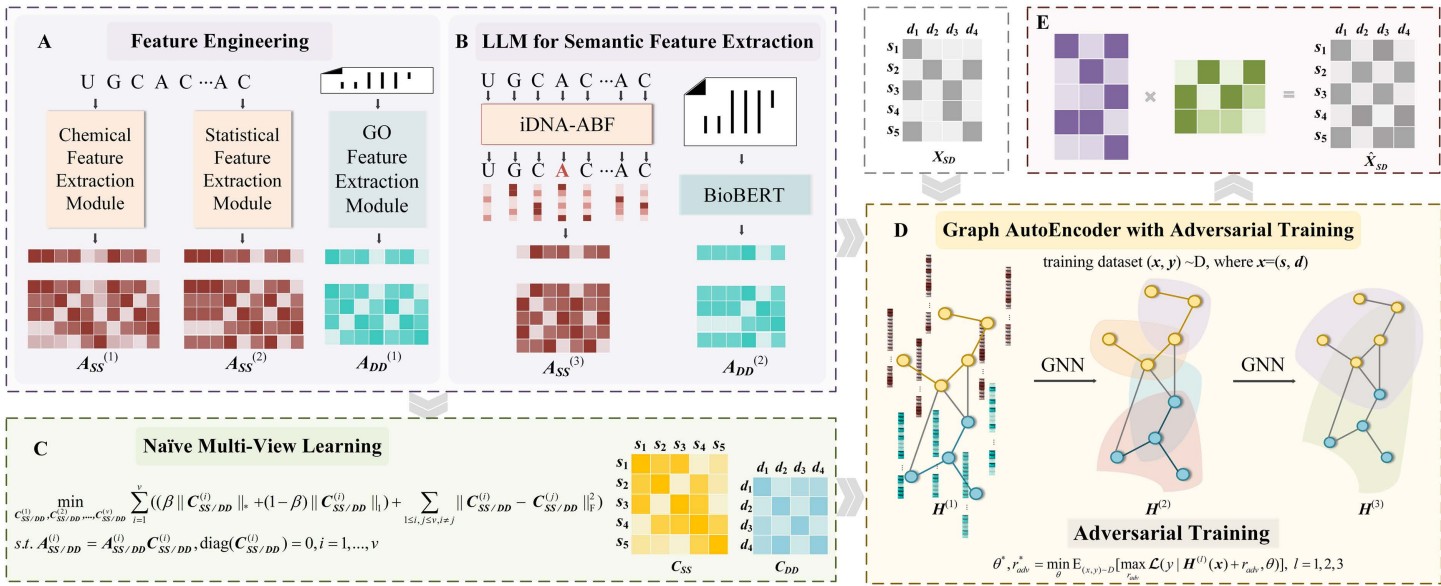

**Fig 1. Framework of PathoRM. A.** iDNA-ABF module to extract contextual information of RM sites. **B.** BioBERT module to extract semantic features, **C.** Naïve Multi-View Learning module to acquire affinity matrices. **D-E** Graph Autoencoder for incorporating prior information and further inference.

the sequence into different scales of biological words, facilitating a transition from natural language to biological language, then automatically and adaptively learn distinguishable semantic features and make relatively accurate predictions for different methylation types in different species. iDNA-ABF exhibits remarkable accuracy across 17 datasets encompassing 3 methylation types, 13 species, and 3 human cell line datasets, demonstrating its capability to extract highly accurate methylation site features. While DNA methylation and RM are different procedures, both rely on the recognition of specific sequence motifs by regulatory proteins or enzymes. This commonality allows the model to transfer its feature extraction capabilities. Thus, as depicted in Fig 1A, we fine-tuned iDNA-ABF on the m6ADA and m7GDA datasets for the task of RM site prediction, enabling the model to derive contextual sequence features associated with RNA modifications.

BioBERT is a biomedical pre-trained language model developed based on BERT, with the aim at capturing the relationships and extracting embeddings from biomedical texts [31]. Trained on huge medical data, BioBERT utilizes multi-layer attention mechanisms to model input sequences at different levels, fully exploring the complex relationships embedded in the disease descriptions. By fine-tuning on tasks such as entity recognition, relation extraction, and knowledge question answering, BioBERT adapts well to various medical tasks, accurately extracting feature embeddings. As Fig 1B illustrates, we initially retrieved relevant pathological descriptions from the PubMed database based on Disease Ontology Identifier (DOID) or Genome-Wide Association Study (GWAS) identifiers, and then directly applied the pre-trained BioBERT model without any further fine-tuning, keeping its parameters frozen throughout.

## 2.4. Naïve multi-view learning algorithm for affinity matrix acquisition

Features extracted from various perspectives reveal a rich landscape of RM sites and diseases. Effectively integrating heterogeneous features is crucial for uncovering unbiased associations among RM sites or diseases in PathoRM. In addition to semantic features extracted from LLM models, we implemented the sequence-derived feature encoding technique outlined in WHISTLE [34] to extract chemical and statistical features for RM sites. Furthermore, we employed the Gene Ontology (GO) feature encoding approach detailed in Wang's method [35] to extract GO features for diseases. Subsequently, we employed a naïve multi-view learning algorithm [32] to capture the complementary and consistent information

from diverse features and finally obtain unbiased affinity matrix. To clearly elucidate the naive multi-view learning algorithm, we initially established the following definitions.

Let $ASS=\{ASS^{(1)}, ASS^{(2)}, ASS^{(3)}\}$ denotes the set of RM feature, $m$ denotes the number of RM sites, and $dim_s^{(i)}$ ($i=1$, 2, 3) denotes the dimension of the RM feature. Then $A_{SS}^{(1)} \in \mathbb{R}^{m \times dim_s^{(1)}}$ is the RM chemical feature matrix with binary 0/1 values, $A_{SS}^{(2)} \in \mathbb{R}^{m \times dim_s^{(2)}}$ represents the RM cumulative nucleotide frequency feature matrix with continuous values ranging from 0 to 1, and $A_{SS}^{(3)} \in \mathbb{R}^{m \times dim_s^{(3)}}$ denotes RM semantic feature matrix with continuous value. Similarly, let $ADD =\{ADD^{(1)}, ADD^{(2)}\}$ denotes the set of disease features, $n$ denotes the number of disease sites, and $dim_d^{(i)}$ ($i=1, 2$) as the dimension of the diseases. Then $A_{DD}^{(1)} \in \mathbb{R}^{m \times dim_d^{(1)}}$ is the disease gene ontology(GO) feature matrix with binary 0/1 values, and $A_{DD}^{(2)} \in \mathbb{R}^{m \times dim_d^{(2)}}$ is the semantic feature matrix of disease with continuous values. $ASS^{(1)}$, $ASS^{(2)}$, $ASS^{(3)}$ describe the RM sites from three different views, while $ADD^{(1)}$, $ADD^{(2)}$ are two aspects for characterizing diseases, which exhibit significant heterogeneity. Naïve multi-view learning algorithm aims to seamlessly integrate these highly heterogeneous multi-view representations into a unified, shared subspace for RM or disease, and finally achieves the RM affinity matrix $CSS$ and disease affinity matrix $CDD$ in a feature-driven and metric-free manner. **Fig 1C** showcases the procedure of utilizing naïve multi-view learning algorithms for acquiring the affinity matrices $CSS$ and $CDD$. Subsequently, we made a detail elaboration about this procedure. For simplicity, we denote $CSS$ or $CDD$ as $C$, and $v$ as the number of views which is corresponds to the types of features. The objective function of the naïve multi-view learning model is outlined as follows.

$$\min_{C^{(1)},C^{(2)},...,C^{(v)}} \sum_{i=1}^{v} ((\beta\|C^{(i)}\|_* + (1-\beta)\|C^{(i)}\|_1) + \sum_{1 \leq i,j \leq v, i \neq j} \|C^{(i)} - C^{(j)}\|_F^2)$$
$$s.t. A^{(i)} = A^{(i)}C^{(i)}, \operatorname{diag}(C^{(i)}) = 0, i = 1, ..., v \tag{1}$$

where $C^{(i)} \in \mathbb{R}^{m \times m}$ is the affinity matrix of the $i$-th view, which reveal the intrinsic interrelationships among RM sites or among diseases. Constrained with low-rankness and sparsity, naïve multi-view learning efficiently diminishes redundancy in molecular features and concurrently extracts complementary information across diverse perspectives. Then $\sum_{i=1}^{v} \sum_{1 \leq i,j \leq v, i \neq j} \|C^{(i)} - C^{(j)}\|_F^2$ is introduced to achieve the consensus representation of RM affinity matrix or disease affinity matrix. The optimization procedure of model (1) can be found in **S1 Text**. After deploying model on the set of RM features, a series of affinity matrices derived from all views, i.e., $CSS^{(1)}$, $CSS^{(2)}$ and $CSS^{(3)}$, can be obtained, and the joint RM affinity matrix $CSS$ is the element-wise average of these affinity matrices. So as for $CDD$.

## 2.5 Graph autoencoder for predicting RM disease associations and prioritizing pathogenic RM Sites

Here, we constructed the RM-disease heterogeneous graph to serve as the input of downstream graph autoencoder. Specifically, S=($s_1$, $s_2$, ..., $s_m$) and D=($d_1$, $d_2$, ..., $d_n$) are regarded as the nodes of the RM-disease heterogenous graph, with edges connecting nodes being determined by validated associations or affinity matrices, where the adjacent matrix of RM-disease association subgraph can be represented as $X_{SD} \in \mathbb{R}^{m \times n}$ and $[XSD]_{ij}$ ($i=1,...,m$; $j=1,...,n$) is set to 1 if ($s_i$, $d_j$) is validated, otherwise 0. Notably, to alleviate noise in the heterogeneous graph, PathoRM sets threshold $\tau$ for $CSS$ and $CDD$, and subsequently defines connections between RM site-RM site as well as disease-disease. A similarity threshold $\tau$ is established, where if the similarity between $s_i$ and $s_j$ exceeds $\tau$, the connection between nodes $s_i$ and $s_j$ is retained; if it falls below $\tau$, the connection between $s_i$ and $s_j$ is removed. The connections between disease nodes follow the same principle. Consequently, the corresponding adjacency matrices are defined as:

$$[\tilde{C}_{SS}]_{ij} = \begin{cases} 1, & [C_{SS}]_{ij} > \tau \\ 0, & [C_{SS}]_{ij} < \tau \end{cases}, [\tilde{C}_{DD}]_{ij} = \begin{cases} 1, & [C_{DD}]_{ij} > \tau \\ 0, & [C_{DD}]_{ij} < \tau \end{cases}$$

Thus, the adjacency matrix of RM-disease heterogeneous graph is defined as:

$$X = \begin{pmatrix} \tilde{C}_{SS} & X_{SD} \\ X_{SD}^T & \tilde{C}_{DD} \end{pmatrix} \in \mathbb{R}^{(m+n)\times(m+n)}$$

Moreover, the semantic features extracted by iDNA-ABF and BioBERT serve as the initial node embeddings of RM site nodes and disease nodes, respectively. The combined node embedding matrix is denoted as $H^{(0)}$, along with $X$ as the inputs of graph autoencoder. PathoRM tries three GNNs, including GCN [36], GraphSAGE [37] and GIN [38], as encoders to propagate the graph embeddings in various message-passing ways by aggregating the prior association and similarity information without changing the graph structures. The message passing and updating rules of GCN, GraphSAGE and GIN are described in the S1 Text. Then the stacking of multiple layers of GNNs can extract high-order features, enhancing the capability of encoder to represent the complex topological structure and node semantic features of the RM-disease heterogeneous graph. However, an excessive number of GNN layers may lead to over-smoothing of the graph, resulting in the loss of crucial structural information and nodes' embeddings. Therefore, in practical operation, the number of GNN layers was set as three, ensuring accurate learning of low-dimensional embeddings while effectively preventing performance degradation caused by over-smoothing. Finally, by deploying the message aggregation and passing methods of GCN, GraphSAGE or GIN, the encoder captures the core semantic and structural information of the RM-disease heterogeneous graph and learns the compressed low-dimensional latent representation $H = [\begin{smallmatrix} H_S \\ H_D \end{smallmatrix}] \in \mathbb{R}^{(m+n)\times k}$, where $H_S \in \mathbb{R}^{m\times k}$ is the latent embedding of RM sites while $H_D \in \mathbb{R}^{n\times k}$ is the latent embedding of diseases. Ultimately, we introduced the matrix decomposition-derived decoder which is also equipped with the activation function σ, to compute the probability of interactions between RM sites and diseases, thereby prioritizing the pathogenic RM sites via reconstructing the RM-disease association matrix. The predicted RM-disease association matrix $\hat{X}_{SD}$ is obtained by the decoder defined in Equation (2):

$$\hat{X}_{SD} = \sigma(H_S^T H_D) \tag{2}$$

## 2.6. Adversarial Training Strategy for Model Optimization

Adversarial learning is a neural network regularization algorithm aimed at improving robustness against small perturbations in inputs. It achieves this by generating adversarial examples—inputs intentionally modified to maximize the model's prediction error—and incorporating them into the training process. Various adversarial training algorithms, including the fast gradient method (FGM) [39], projected gradient descent (PGD) [40], and fast gradient sign method (FGSM) [41] have been proposed to enhance performance in tasks such as image recognition, object detection, and natural language processing. Rooted in the FGM algorithm, we introduced adversarial training to the graph autoencoder, ultimately enabling it to generate more robust latent embeddings. Specifically, let $\mathbf{L}(\mathbf{y} \mid H^{(l)}(\mathbf{x}), \theta)$ denote the loss function of the $i$-th ($i = 1,2,3$) layer of GNN encoder, where $(\mathbf{x}, \mathbf{y})$ represents the training data, $\theta$ is the model parameter, and $H^{(l)}$ denotes the feature embedding of GNN encoder at the $l$-th layer. As (3) shows, the optimal parameters of PathoRM can been achieved via backpropagation procedure without adversarial training algorithm.

$$\theta^* = \arg\min_{\theta} \mathbb{E}_{(x,y)\sim D} \mathcal{L}(y|H^{(l)}(x), \theta), l = 1, ..., L \tag{3}$$

Rooted in FGM algorithm, we introduced adversarial perturbation $r_{adv}$ to the GNN embeddings. Consequently, the optimization procedure of optimal perturbation $r_{adv}^*$ and optimal parameter $\theta^*$ can be cast into the following max-min problem:

$$\theta^*, r_{adv}^* = \min_{\theta} \mathbb{E}_{(x,y)\sim D}[\max_{r_{adv}} \mathsf{L}(y|H^{(l)}(x) + r_{adv}, \theta)], l = 1, 2, 3 \tag{4}$$

At each training step, a two-stage procedure is adopted to get the worst-case perturbation and optimal model parameter. Firstly, we temporarily set $\theta$ as $\hat{\theta}$, and obtained $r^*_{adv}$ through the following formula.

$$r^*_{adv} = \arg\max_{r_{adv}} \mathsf{L}(y|H^{(l)}(x) + r_{adv}, \theta)], l = 1, 2, 3 \tag{5}$$

(5) cannot be directly solved via computing its derivative with respect to $r_{adv}$. Thus, we proceeded to expand $\mathsf{L}(y|H^{(l)}(x) + r_{adv}, \theta)$ using Taylor series.

$$\mathsf{L}(y|H^{(l)} + r_{adv}, \hat{\theta})] = \mathsf{L}(y|H^{(l)}, \hat{\theta})] + r_{adv}\mathsf{L}(y|H^{(l)}, \hat{\theta})], l = 1, \dots, L \tag{6}$$

where $||r_{adv}||_2 \le \varepsilon$.

Since the adversarial perturbation in essence results in an increase in the loss value, maximizing $\mathsf{L}(y|H^{(l)} + r_{adv}, \hat{\theta})$ is equivalent to maximizing $r_{adv}\mathsf{L}(y|H^{(l)}, \hat{\theta})$. Hence, we obtain optimal adversarial perturbation which is shown as (7).

$$r^*_{adv} = \frac{\nabla_{H^{(l)}}\mathsf{L}(y|H^{(l)}, \hat{\theta})}{||\nabla_{H^{(l)}}\mathsf{L}(y|H^{(l)}, \hat{\theta})||_2}, l = 1, 2, 3 \tag{7}$$

Finally, setting $r_{adv}$ as $r^*_{adv}$, we solve the minimization problem (8) to obtain the optimal parameters $\theta^*$.

$$\theta^* = \min_{\theta} \mathsf{E}_{l=1,2,3}\mathcal{L}(y|H^{(l)} + r^*_{adv}, \theta)] \tag{8}$$

### 2.7. "Guilty-by-Association"-derived Negative Sampling Scheme for Model Training

High-quality negative samples are crucial for improving the training efficiency and predictive performance of deep learning models. Here, we proposed a negative sample selection method based on "guilty-by-association" principle, guiding the model to select reliable negative samples from candidate samples. Specifically, taking $(s_i, d_j)$ as an example. S represents the set of RM sites, and S\$s_i$ represents the set of other RM sites which except for $s_i$. $s_k$ is the least similar site to $s_i$ and simultaneously $(s_k, d_j)$ is a candidate sample. The principle of guilty-by-association posits that RM sites with higher similarity are more likely to be involved in the same pathways of disease occurrence and progression. Conversely, RM sites with lower similarity are less likely to be involved in the same disease pathways. Based on this premise, $(s_k, d_j)$ can be defined as negative samples.

## 3. Results

To validate the predictive power of our model, we conducted a comprehensive comparison of PathoRM against leading baseline methods from three distinct perspectives: (i) overall performance on a balanced dataset utilizing 10-fold cross-validation, (ii) overall performance on an imbalanced dataset utilizing 10-fold cross-validation(10-fold CV), and (iii) performance in Leave-One-Disease-Out-Cross-Validation (LODOCV) setting. While numerous convex optimization-based models have been developed to address RM-disease association prediction, deep learning-based models remain scarce. Therefore, for a thorough and fair evaluation, we selected three widely used deep learning models—DeepDTA [23], Deep-DCA [24], DeepConv-DTI [42]—and one multi-view learning-based model—AMVML [43]—for comparison.

### 3.1. Experimental design

To systematically assess the effectiveness of PathoRM in prioritizing pathogenic RM sites, we performed PathoRM and its competing models in two experimented settings, including 10-fold CV and LODOCV, which the predictive capabilities

of the models under both "warm-star" and "cold-star" scenarios. In 10-fold CV scheme, all known RM-disease association samples, referred to as positive samples, are divided into roughly equal-sized ten subsets. In each fold, nine subsets are designated as training sets, while the remaining subset containing positive samples is merged with samples selected through a negative sampling strategy from the candidate pool to compose the test set. Additionally, we constructed balanced and imbalanced scenarios to comprehensively evaluate the predictive performance and robustness of the algorithms. In terms of the balanced scenario, the number of selected negative samples equals the number of positive samples, while in the imbalanced scenario, the number of selected negative samples is ten times that of the positive samples. Under both settings, the performance of the models on balanced and imbalanced test sets is comprehensively evaluated using ROC curves, PR curves, AUC, AUPR, ACC, Precision, Recall, Specificity, and F1 score. The calculation for these metrics can be found in the attached S1 Text. LODOCV was designed to assess the models' capabilities in prioritizing the pathogenic RM sites under the "cold-start" scenario. Briefly, taking disease $d_i$ as an example, the remaining diseases constitute the set D\$d_i$. Then $d_i$-associated validated samples $(s, d_i)$, $s \in S$, are regarded as test sample while validated $s$-D\ $d_i$ pairs are regarded as training set. In the training procedure, PathoRM takes full advantages of prior similarity information as well as validated association information, and explore the latent embeddings in the heterogeneous RM-disease space, then obtained the predicted association scores of samples $(s, d_i)$, $s \in S$. The AUC metric was applied to evaluate PathoRM and its competing models under the LOODOCV scheme. By deploying each model on the disease set D=$(d_1, d_2, …, d_n)$ with the LODOCV scheme, $n$ AUC values can be obtained. Finally, we accessed the models' ability in "cold-start" scenario by comparing the distribution of AUC values of PathoRM and its competing models.

### 3.2. Comparison methods

**DeepDTA** is a classical deep learning model in predicting the affinity between drugs and targets. To enhance its applicability in predicting pathogenic RM sites, DeepDTA partitions RM host sequences into 3-mers, representing them as a sequence of "biological words", and subsequently encodes them. Similarly, disease text sequences are encoded after constructing a dictionary of disease text descriptions. This pre-processing step aims to derive effective representations of RM sites and disease information, thus providing suitable inputs to the model. Following this, DeepDTA employs two CNN modules to learn feature embeddings of RM sites and diseases. Each CNN block consists of three consecutive convolutional layers followed by pooling layers for down-sampling. Ultimately, the features from these two pooling layers are concatenated and fed into three fully connected layers to generate final prediction results.

   **DeepDCA** is an extension of DeepDTA, which is also proposed for predicting the affinity between drugs and targets. Building upon the same encoding methods as DeepDTA, DeepDCA combines CNN and LSTM architectures to extract local feature embeddings of RM sites and diseases, respectively. Additionally, DeepDCA introduces a bidirectional attention mechanism to model the interaction between RM sites and diseases, thereby improving the model's representation of drug-target relationships and ultimately yielding prediction results.

   **DeepConv-DTI** is also dedicated to predicting drug-target interaction. In the context of RM-disease association prediction, it adopts the encoding approach for RM site host sequences and disease text from DeepDTA, utilizing a fully connected neural network to learn feature representations of RM site host sequences, and employing CNN and pooling layers to capture semantic features of disease text. Subsequently, DeepConv-DTI concatenates extracted drug features with protein features and passes them through a fully connected neural network to produce prediction results.

   **AMVML** is a multi-view learning-based model which once used to predict the miRNA-disease associations. In this study, AMVML is retrained and applied to predict pathogenic RM sites. Specifically, AMVML treats similarity calculated under different features of RM sites and diseases as distinct views. It then employs a multi-view subspace learning algorithm to fuse various RM site/disease similarity information and known RM site-disease associations, resulting in a completed RM site-disease association matrix.

## 3.3. Implementation and optimization

We performed all experiments with PyTorch 1.13.1 on NVIDIA A100 GPU. Xavier initialization strategy [44] was used to prevent gradients from exploding and vanishing in the consequent GNN layers. Then the Adam optimizer was used in the model training procedure, where weighted cross entropy loss $\mathcal{L}_{\text{WCE}}$ was applied to quantify the difference between the predicted outputs and true target values and simultaneously to mitigate the issues caused by imbalanced data.

$$\mathcal{L}_{\text{WCE}} = -\frac{1}{m \times n}\left(\frac{|\Omega^-|}{|\Omega^+|}\sum_{(s_i,d_j)\in\Omega^+}\log[\hat{\boldsymbol{X}}_{\boldsymbol{SD}}]_{ij} + \sum_{(s_i,d_j)\in\Omega^-}\log(1-[\hat{\boldsymbol{X}}_{\boldsymbol{SD}}]_{ij})\right) \tag{9}$$

where $\Omega^+$ and $\Omega^-$ denote the set of positive instances and the set of negative instances, respectively, while $|\Omega^+|$ and $|\Omega^-|$ denote the numbers of positive and negative instances, respectively.

## 3.4. PathoRM shows the best prediction performance in 10-fold CV experiments across datasets

Here, we compared PathoRM with its competing methods on both balanced and imbalanced m6ADA datasets under 10-fold CV scheme. For fair comparison, the models were executed and evaluated with the same training sets and test sets which were split by 10-fold CV. Since disease trait description is a unique data type, different from drug and target data which can be directly encoded by sequence elements, the three comparative models use a combination of GO features and embeddings generated by BioBERT as input embeddings for diseases. To align with PathoRM, the comparison methods concatenate RM chemical features, statistical features, and embeddings generated by iDNA-ABF as input embeddings for RM sites.

The performance was mainly measured in terms of AUC and AUPR, following by other measurements F1 score, ACC, Recall, Specificity and Precision. Fig 2A–E illustrates the overall comparison performance between PathoRM and its competing methods on four balanced datasets under 10-fold CV. As illustrated in **Fig 2AB**, PathoRM showcases promising performance on both the balanced and imbalanced m6ADA datasets Specifically, PathoRM achieves an AUC of 0.9878±0.0195 and an AUPR of 0.9904±0.0295. In terms of mean AUC, PathoRM leads a large margin of DeepConv-DTI, DeepDTA, DeepDCA, and AMVML by 23.65%, 26.43%, 9.89%, and 7.86%, respectively. Similarly, in terms of AUPR, PathoRM exhibits a superiority of 27.44%, 31.39%, 35.55%, and 10.05% over its comparative counterparts. Notably, its superiority becomes even more clear on the imbalanced m6ADA dataset, with the highest mean AUC of 0.9737 and mean AUPR of 0.8815, leading a large margin from 17.74% to 73.12%, thus showcasing robust capability in the classification of m 6A-disease associations.

What's more, AMVML emerges as the second-best performer on the m6ADA dataset, falling slightly short of PathoRM. Specifically, on the balanced dataset, AMVML achieves an AUC of 0.9092±0.0204 and an AUPR of 0.8899±0.0314, while on the imbalanced dataset, it records an AUC of 0.9388±0.0120 and an AUPR of 0.7041±0.0599. In contrast, the three natural language processing algorithms—DeepConv-DTI, DeepDTA, and DeepDCA exhibit relatively poor performance. **Fig 2CD** further demonstrate the superiority of PathoRM by comparing it with DeepConv-DTI, DeepDTA, DeepDCA and AMVML in terms of F1 score, Precision, Recall, ACC and Specificity on m6ADA. Regardless of balanced or imbalanced situation, PathoRM outperforms the comparative algorithms across all metrics and achieves a fine balance. Simultaneously, increasing the negative samples ninefold resulted in a slight decline in AUPR, F1 score, Precision, and Recall for both PathoRM and AMVLV, while leading to a drastic drop in these metrics for DeepDTA, DeepDCA, and DeepConv-DTI. This underscores the high predictive accuracy and robustness of PathoRM and AMVLV.

To validate the robustness of PathoRM across datasets, further comparisons were made by examining the performance of each algorithm on the m7GDA dataset. S1 Table presents the results of 10-fold cross-validation for PathoRM and its comparative algorithms on m7GDA dataset. PathoRM consistently demonstrates superior performance, particularly in

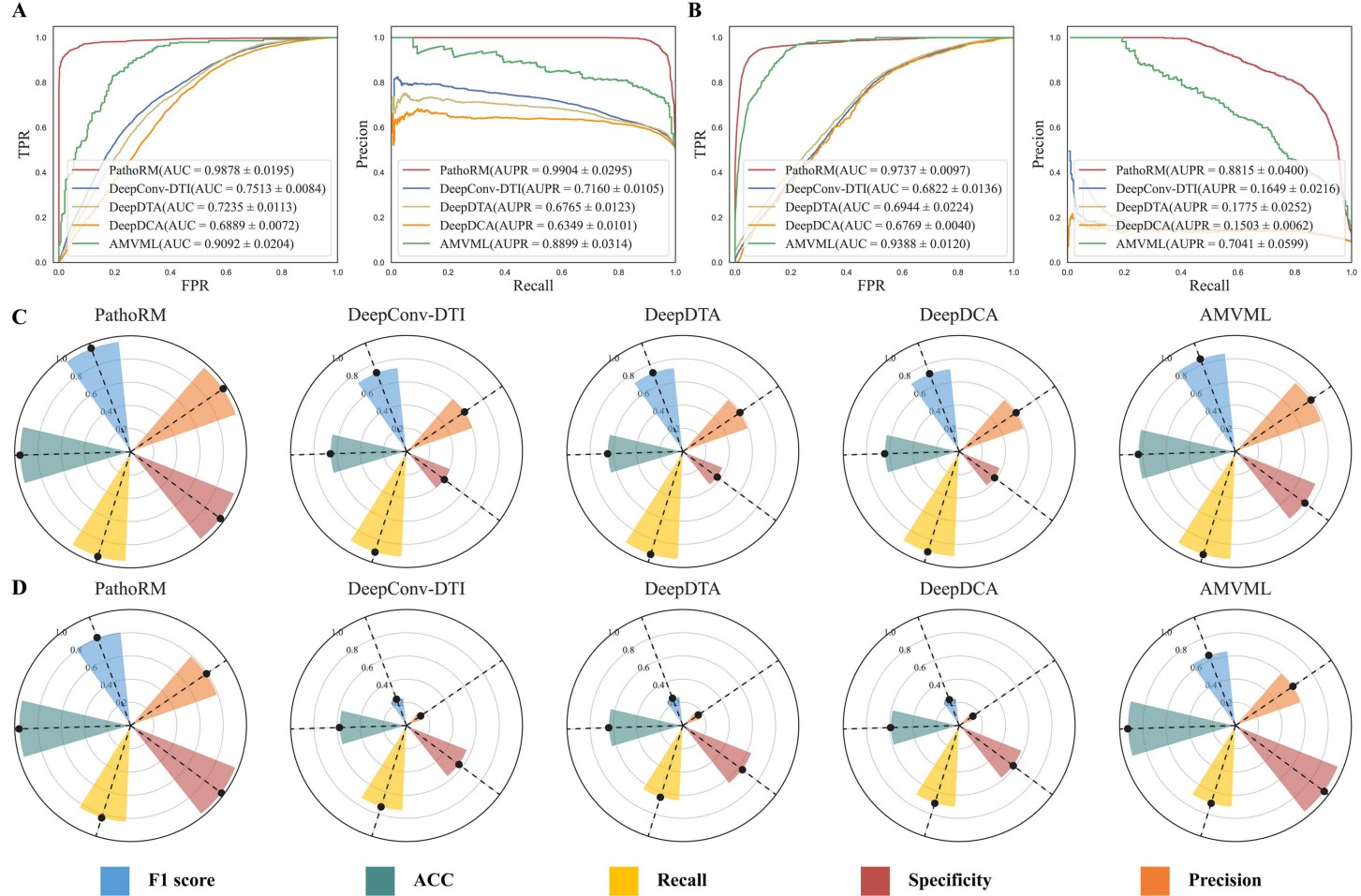

**Fig 2. Overall performance comparison results between PathoRM and its competing methods on m6ADA dataset with 10-fold CV. A**. The ROC and PR curves of PathoRM and its competing methods on balanced setting. **B**. The ROC and PR curves of PathoRM and its competing methods on imbalanced setting. **C**. The performance of PathoRM, DeepConv-DTI, DeepDTA, DeepDCA and AMVML in terms of ACC, Precision, Recall, F1 score, and Specificity on balanced setting. **D**. The performance of PathoRM, DeepConv-DTI, DeepDTA, DeepDCA and AMVML in terms of ACC, Precision, Recall, F1 score, and Specificity on imbalanced setting.

terms of AUC, AUPR, and Recall, showcasing robust predictive capabilities for RM-disease associations and the identification of positive samples. Additionally, we also compared PathoRM with traditional machine learning models, including Support Vector Machine (SVM), Random Forest (RF), and Gradient Boosting Machine (GBM), which rely on handcrafted features and simpler architectures. The results can be found in S1 Fig, and the parameterization analysis of PathoRM can be found in S1 Text and S2 Table.

The statistical significance of the performance differences between PathoRM and its competing methods in the 10-fold CV scheme was evaluated using the Wilcoxon test, with the $p$ value indicating the level of significance. As **Fig 3** shows, regardless of m6ADA and m7GDA dataset, $p$ values between PathoRM and its comparison methods are far less than 0.05, which further indicates the superiority of PathoRM is statistically significant. To this end, the results manifest that our proposed PathoRM significantly outperforms the leading baselines across all datasets, metrics and scenarios. It can be inferred that the outstanding performance of PathoRM may be attributed to the utilization of multi-view algorithms, which

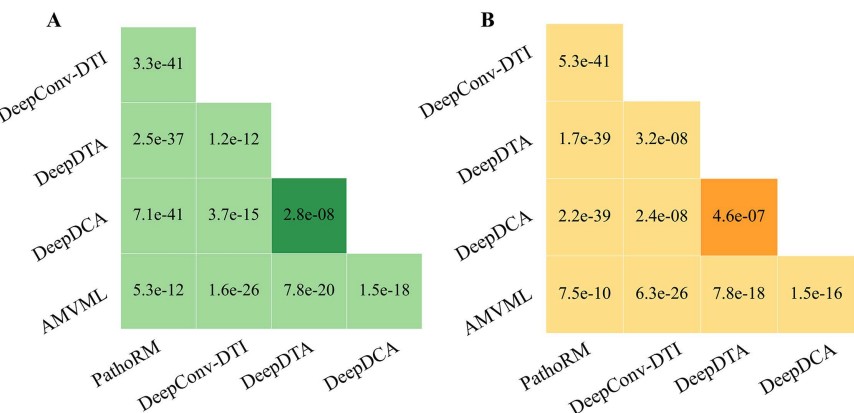

Computational
Biology

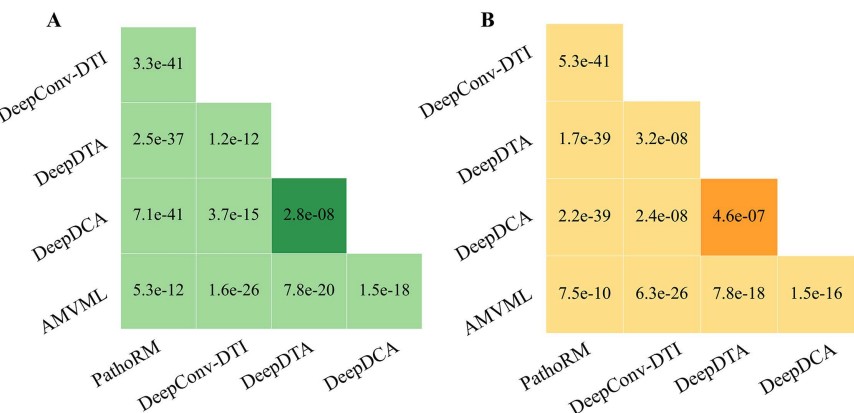

**Fig 3. Statistical significance of differences(*p* value) among PathoRM, DeepCon-DTI, DeepDTA, DeepDCA and AMVML on A. m6ADA dataset B. m7GDA dataset.**

integrate various features of RM sites and diseases, thereby reducing the model's reliance on specific features. Additionally, it learns consistent complementary information among various features, enabling the capture of unbiased associations. PathoRM extends multi-view learning into the framework of deep learning, then utilizing affinity matrix to initialize the edges of the RM-disease heterogenous graph, thus fully leveraging the prior information.

To intuitively explain why our proposed PathoRM performs better than the other approaches, we further visualized the distribution of feature embedding space of PathoRM, DeepConv-DTI, DeepDTA and DeepDCA in all test sets generated by 10-fold CV. **Fig 4** show the t-distributed Stochastic Neighbor Embedding (t-SNE, a widely used tool for visualizing high-dimensional data by giving a 2D map) [45] visualization results of them, where red dots represent the positive samples while green dots represent the negative samples. As seen from **Fig 4**, the feature space of positive associations and that of negative associations are relatively discriminative, well-separated and clear. Additionally, the samples within one cluster are closely compact rather than disperse. As comparison, the samples of feature space generated by DeepConv-DTI is slightly mixed. The worse situation occurs in DeepDCA and DeepDTA where their feature spaces are connected and totally mixed with each other, and it is hard to circle the boundary for each class. The t-SNE performance align perfectly with the comparison results shown in **Fig 3**. To this end, we can attribute the superiority of PathoRM to its capability in accurately recognizing and grasping the underlying feature patterns of positive and negative RM-disease associations.

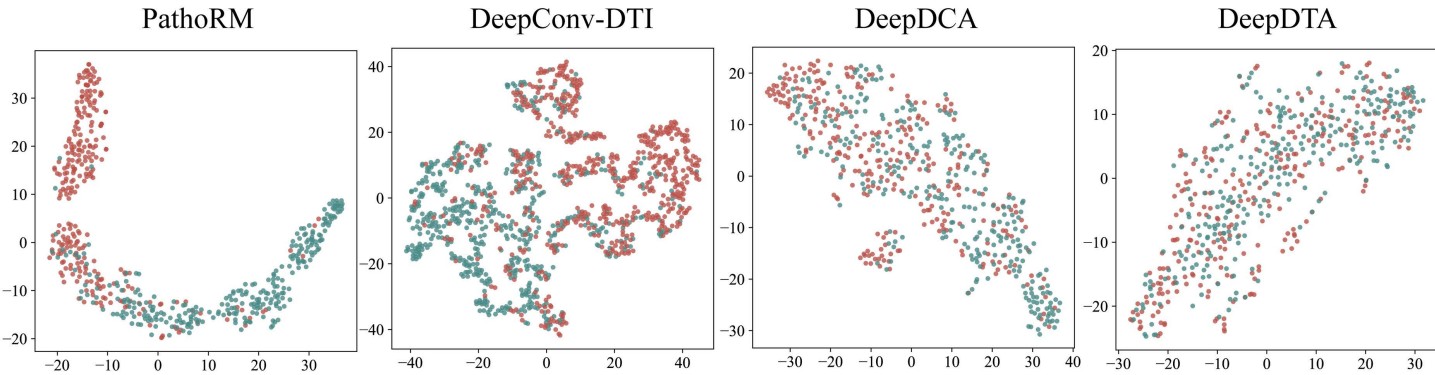

**Fig 4. Feature space distribution of PathoRM, DeepConv-DTI, DeepDTA and DeepDCA with t-SNE.**

### 3.5. PathoRM Outperforms the State-of-the-art Methods in the LODOCV Experiments

To further evaluate the capabilities of PathoRM and its competing methods in a "cold-start" scenario, we conducted LODOCV experiments on them. **Fig 5AB** display the distribution of AUC statistic for PathoRM, DeepConv-DTI, Deep-DTA, DeepDCA, and AMVLM on the m6ADA and m7GDA datasets in the LODOCV experiments, respectively. As **Fig 5A** shows, on the m6ADA dataset, PathoRM generated 2860 AUC values that exhibited a right-skewed distribution, where the AUC values are highly concentrated around 0.95, indicating nearly accurate prediction of pathogenic RM sites. The AUC values produced by AMVML also displayed a right-skewed distribution, centered at 0.85. However, the AUC values generated by DeepDTA and DeepDCA were concentrated between [0.6, 0.8], performing less promising compared to PathoRM and AMVML. The AUC distribution for DeepConv-DTI ranged between [0.5, 0.7], indicating the poorest performance among the methods. While as **Fig 5B** shows, the peaks of AUC scores for DeepDTA, DeepConv-DTI, DeepDCA, AMVML and PathoRM successively shifted to the right on the m7GDA dataset. Furthermore, PathoRM generates 786 AUC values with the smallest variance, concentrated between [0.8, 1.0], consistently exhibiting the best performance among all methods. This further demonstrates the robustness and applicability of PathoRM in prioritizing relevant RM sites for diseases.'

### 3.6 GIN Achieves the Best Performance Among Diverse GNNs

To further investigate the effectiveness of diverse GNN variants, including GIN, GCN and GraphSAGE, we implemented the graph autoencoder of PathoRM with GNN variants on m6ADA dataset. **Fig 6A** shows their performance which is evaluated by AUC and AUPR. In general, GIN has the leading performance equipped with the highest mean AUC/AUPR and relatively low standard variance (balanced dataset: AUC = 0.9878 ± 0.0050, AUPR = 0.9905 ± 0.0033; imbalanced dataset: AUC = 0.9737 ± 0.0102, AUPR = 0.8815 ± 0.042), while GCN shows the lowest mean AUC/AUPR values with the largest variances (balanced dataset: AUC = 0.9660 ± 0.0128, AUPR = 0.9662 ± 0.0138; imbalanced dataset: AUC = 0.9477 ± 0.0127, AUPR = 0.7779 ± 0.0632). Therefore, taking GIN as the graph encoder significantly enhances the algorithm's prediction accuracy. The results may attribute to the message passing methods of GIN, GraphSAGE and GCN. Specifically, GIN is designed based on graph isomorphism principles. During message passing, the connectivity of the RM site-disease heterogeneous graph and the graph structure remain unchanged [38]. Coincidently, the RM-disease heterogeneous graphs in different GNN layers are isomorphic. Moreover, GIN iteratively aggregates information from neighbouring nodes in

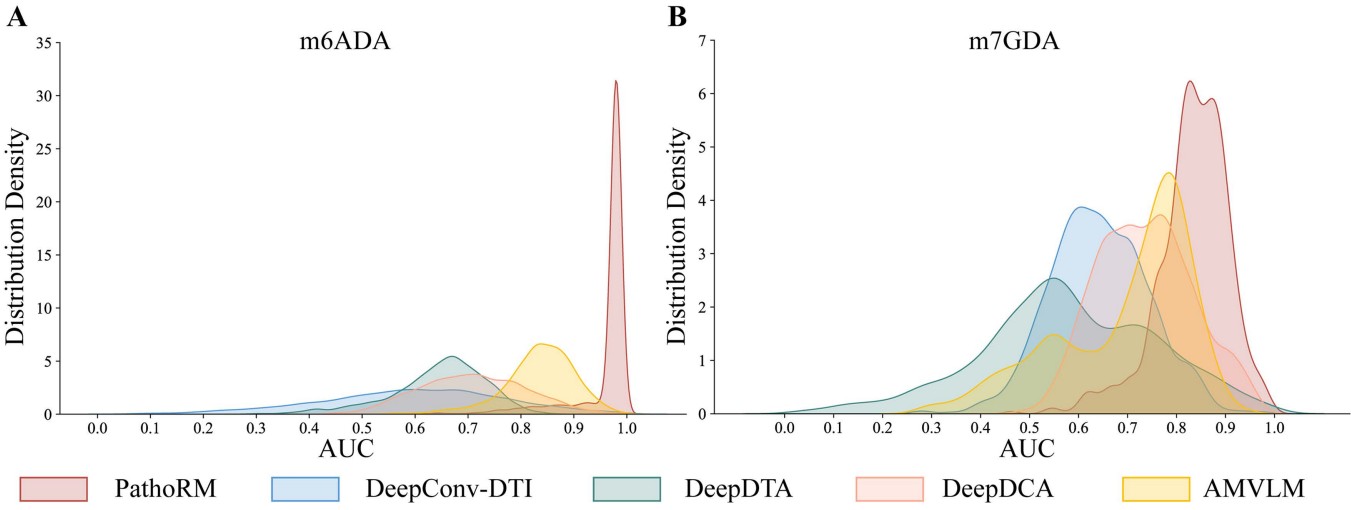

**Fig 5. Distritbuions of AUC statistic for PathoRM, DeepConv-DTI, DeepDTA, DeepDCA and AMVLM under LODOCV scheme. A.** m6ADA dataset **B.** m7GDA dataset.

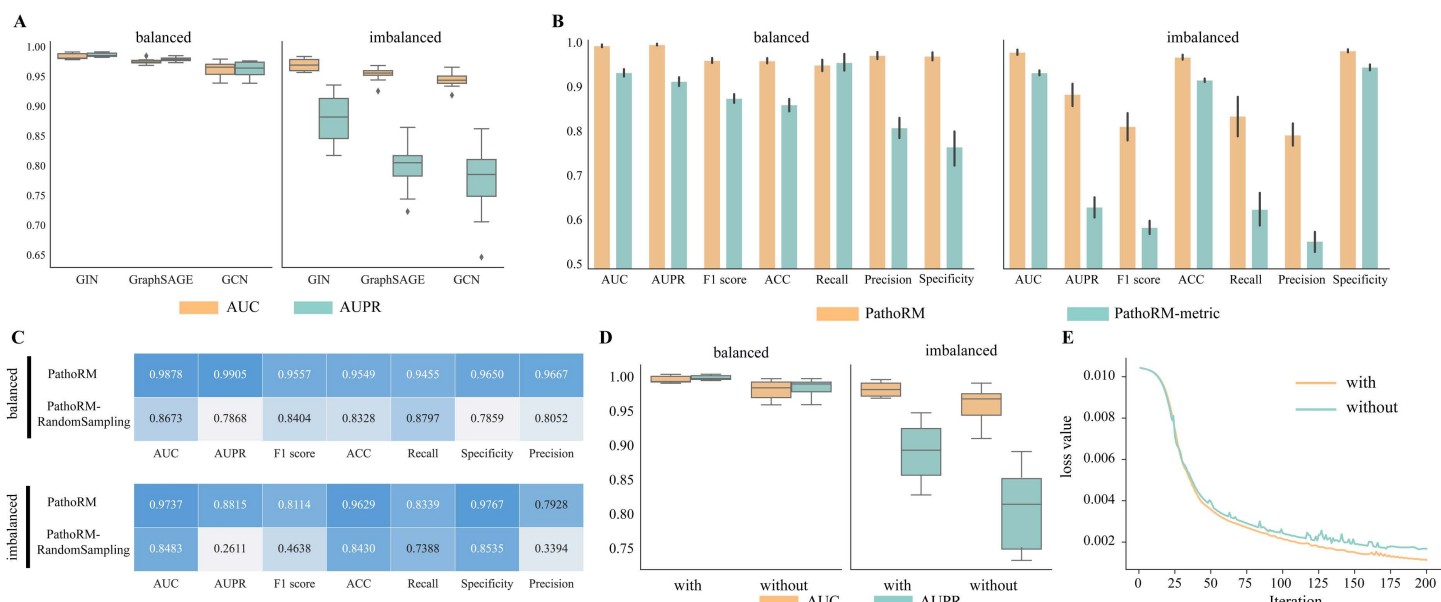

**Fig 6. Performance of ablation studies. A** Performance comparison of PathoRM with GIN, GraphSAGE and GCN. **B** Performance comparison of PathoRM and PathoRM-metric. **C** Performance comparison of PathoRM and PathoRM-RandomSampling. **D** Performance comparison of PathoRM with and without adversarial training scheme. **E** Loss curves generated by PathoRM with and without adversarial training.

multiple layers of graph isomorphism layers, capturing global information through pooling operations. This enables GIN to consider the overall structure more comprehensively when processing graph data, facilitating a more holistic understanding of relationships within the graph. On the other hand, GraphSAGE and GCN are not specifically designed for isomorphic graphs, and their information aggregation processes are more sensitive. Specifically, GCN aggregates embeddings in a simple manner by averaging embeddings of all features of neighbour nodes [36]. GraphSAGE first samples its neighbours and then updates feature embeddings by integrating the current embedding with aggregated neighbour information [37]. The averaging approach may introduce noise, while random neighbour sampling can mitigate this issue. Therefore, GIN exhibits optimal performance in this context. Additionally, GraphSAGE achieves better prediction performance than GCN due to its more reasonable neighbour aggregation method.

### 3.7. Multi-view learning is optimal for unbiased similarity information acquisition

Here, we further investigated the contribution of multi-view learning stage in PathoRM by replacing it with similarity metrics while keeping the other components unchanged. Given various features exhibit diverse data structures, different metrics are employed to compute their similarities. In terms of the RM features, $ASS^{(1)}$ denotes binary-encoded representations of chemical structures, while $ASS^{(2)}$ signifies cumulative nucleotide frequencies encoded as continuous values ranging from 0 to 1. Thus, we applied the Jaccard coefficient and Cosine coefficient on $ASS^{(1)}$ and $ASS^{(2)}$ to achieve $\tilde{X}_{SS}^{(1)}$ and $\tilde{X}_{SS}^{(2)}$, respectively. Then $ASS^{(3)}$ represents the semantic features extracted by iDNA-ABF, which is encoded with continuous numerical values. Therefore, we still leveraged the Cosine coefficients on $ASS^{(3)}$ to derive semantic similarity $\tilde{X}_{SS}^{(3)}$ from it. The same process applies to the set of disease features to achieve $\tilde{X}_{DD}^{(1)}$ and $\tilde{X}_{DD}^{(2)}$. Finally, element-mean operation was applied on $\tilde{X}_{SS}^{(1)}$, $\tilde{X}_{SS}^{(2)}$ and $\tilde{X}_{SS}^{(3)}$ for taking each view into consideration and obtaining the comprehensive RM similarity $\tilde{X}_{SS}$, so as for disease similarity $\tilde{X}_{DD}$. Based on them, we reconstructed heterogeneous graph of known associations between RM sites and diseases and the other components remain unchanged. This similarity-dependent model is denoted as

PathoRM-metric. Fig 6B illustrates the performance of PathoRM and PathoRM-metric across scenarios and metrics on m6ADA dataset.

As **Fig 6B** shows, compared with PathoRM, PathoRM-metric demonstrates a decline in performance across all scenarios and metrics. Particularly in imbalanced datasets, a substantial decrease of approximately 30% is observed in AUPR, F1 score, Recall, and Precision metrics. This strongly indicates the significant role of employing multi-view subspace learning algorithms in acquiring unbiased RM site/disease similarities, reducing noise, and enhancing predictive performance. Specifically, multi-view subspace learning enables the model to automatically select and integrate features from different views, thereby learning RM site-RM site and disease-disease similarities in a feature-driven manner. In contrast, RM site/disease similarities obtained through similarity metric approaches may fail to accurately learn the true relationships between RM sites and diseases since the calculation process itself may potentially introduce some degree of noise. Hence, multi-view subspace learning is vital for constructing unbiased RM site-disease heterogeneous graphs and effectively capturing association patterns with graph autoencoders.

### 3.8. Negative sampling scheme enables the promising prediction performance of pathoRM

To assess the efficacy of the proposed "guilty-by-association"-derived negative sampling scheme, we employed random sampling to select negative samples and then retrained PathoRM, which is denoted as PathoRM-RandomSampling. **Fig 6C** illustrates the performance comparison between PathoRM and PathoRM-RandomSampling on m6ADA dataset, where darker shades represent higher scores. As **Fig 6C** shows, on the balanced dataset, PathoRM consistently outperforms PathoRM-RandomSampling across all metrics, showcasing a significant 20.37% advantage in AUPR. Furthermore, the advantages stemming from the "guilty-by-association"-derived negative sampling strategy are underscored by a notable 50% increase in AUPR attained by PathoRM over PathoRM-RandomSampling on the imbalanced dataset. This highlights the effectiveness of the "guilty-by-association"-derived negative sampling method in capturing distinctive feature patterns of both positive and negative samples. Consequently, it enhances model training efficiency and accuracy while mitigating the risk of overfitting to noise and redundant information.

### 3.9. Adversarial training scheme enhances the robustness of PathoRM

Adversarial training is another strategy we used in the training process of PathoRM. Here, we analysed its contribution to the prediction performance by retraining PathoRM without utilizing adversarial training. Taking m6ADA dataset as an example, **Fig 6D** shows the comparison results of PathoRM with and without adversarial training under 10-fold CV experiments. Specifically, model with adversarial learning achieves higher mean AUC and AUPR values, along with exhibiting smaller variances, indicating greater robustness of the model. Then we further investigated the reasons behind this phenomenon. **Fig 6E** depicts the curves of loss function of PathoRM with and without adversarial learning during the training process. PathoRM with adversarial training demonstrates lower and more stable training losses, whereas PathoRM without adversarial training exhibits higher losses with unstable local fluctuations. This suggests that adversarial learning, by introducing adversarial noise at the GNN layer, enables the model to learn resistance to minor perturbations during training process, thereby mitigating overfitting on the test set and enhancing the robustness and generalization of PathoRM.

### 3.10. PathoRM Associates pathology with conserved Motif

RM sites with identical motifs in their host sequences may execute similar biological functions and participate in the same disease processes. For better interpreting model's decision at the sequence-level, PathoRM outputs the normalized site importance scores through attention mechanisms of iDNA-ABF without any prior base-level annotations. First, we investigated the impact of RM sequence length on PathoRM's performance and identified the optimal sequence for subsequent

motif identification. Specifically, we systematically truncated the original m⁷G-centered (41 bp) to 21 bp and 11 bp, the original m⁶A-centered (65 bp) to 35 bp and 15 bp, while preserving site-centric alignment. These variants were incorporated into PathoRM and assessed via 10-fold CV, with AUC and AUPR comparisons detailed in S2 Fig in S1 Text. Based on the results, we selected the 41 bp m⁷G-centered sequence and the 65 bp m⁶A-centered sequence for further analysis. Then, all validated associations $(s, d) \in \Omega$ are regarded as positive training samples while the others $(s, d) \notin \Omega$ are taken as candidate samples. By applying the well-trained PathoRM on the training set, all candidate RM sites associated with specific disease are ranked by the predicted scores. Here, we took breast cancer and Alzheimer's disease as examples, and picked up the top three sites for convincing results. Then for granting insights into how PathoRM makes determinations at the base-level, we extracted the attention matrix generated by iDNA-ABF, which measures the correlations among all residues. For each site, we calculated its contribution by summing its correlations with the remaining sites, and normalized it through a min-max scaling process. A higher normalized value indicates stronger dependency with the upstream and downstream regions, thus demonstrating the important role this site plays in the determination process of PathoRM. Fig 7AB respectively depict the distribution of nucleotide importance scores for the top three m⁷G host sequences which are most associated with breast cancer and Alzheimer's disease. To investigate the biological meaning of the identified regions with high scores, we further utilized STREME [46] to identify conserved motifs within known breast cancer/Alzheimer's disease-related m⁷G host sequences. Fig 7CD respectively display the most significantly enriched motifs in m⁷G host sequences associated with breast cancer and Alzheimer's disease.

As Fig 7 shows, conserved motif in breast cancer-related m⁷G host sequences is "CCCGCCU" with each base being equally enriched. Considering the distribution of base importance scores shown in Fig 7A, it can be observed that the high-probability distribution region largely overlaps with the conserved motif in Fig 7C. The conserved motif in Alzheimer's disease-related m⁷G host sequences is "AAGUGGG" with a higher enrichment frequency of GU, indicating greater conservation. The high-probability distribution region shown in Fig 7B essentially overlaps with the motif "AAGUGGG" in Fig 7D with the highest importance scores observed at the GU positions. This phenomenon suggests that the attention matrix can largely reflect the conserved motifs of disease-related RM host sequences. These conserved motifs may function as regulatory elements or correspond to conserved functional domains in RNA secondary structures, playing crucial biological roles in the onset and progression of diseases. For instance, the GU-enriched regions are typically associated with snRNA and intron splicing processes. The GU-enriched region is recognized by the spliceosome, a complex of RNA and protein molecules responsible for splicing, and it serves as an essential signal for the initiation of splicing [47]. Therefore, despite PathoRM not being explicitly designed to identify these conserved motifs, post-training analysis of the model architecture through attention mechanisms enables the recognition of upstream and downstream conserved patterns associated with specific diseases.

## 4. Discussion

Herein, we developed a deep learning model PathoRM for RM site-disease association prediction. Equipped with large language model, multi-view learning, graph autoencoder and training strategies, PathoRM has the leading performance when compared with widely used deep learning-based models, multi-view learning-based models and various traditional machine learning methods under both "warm-start" and "cold-start" scenarios. Additionally, t-SNE analysis results indicate that outstanding performance of PathoRM is attributed to its excellent ability to recognize the distinctive feature patterns of positive and negative association samples. Additionally, ablation studies approved the rationale of model design and the effectiveness of each component. Notably, even without explicit annotations for sites, the present pipeline has the potential to capture the intrinsic pathogenic regions, which is overlapped with the conserved motif, in the RM host sequence with the attention mechanism, offering biological insights into the decision-making procedure. Thus, PathoRM could emerge as a promising tool for predicting RM-disease associations and inferring conserved motif.

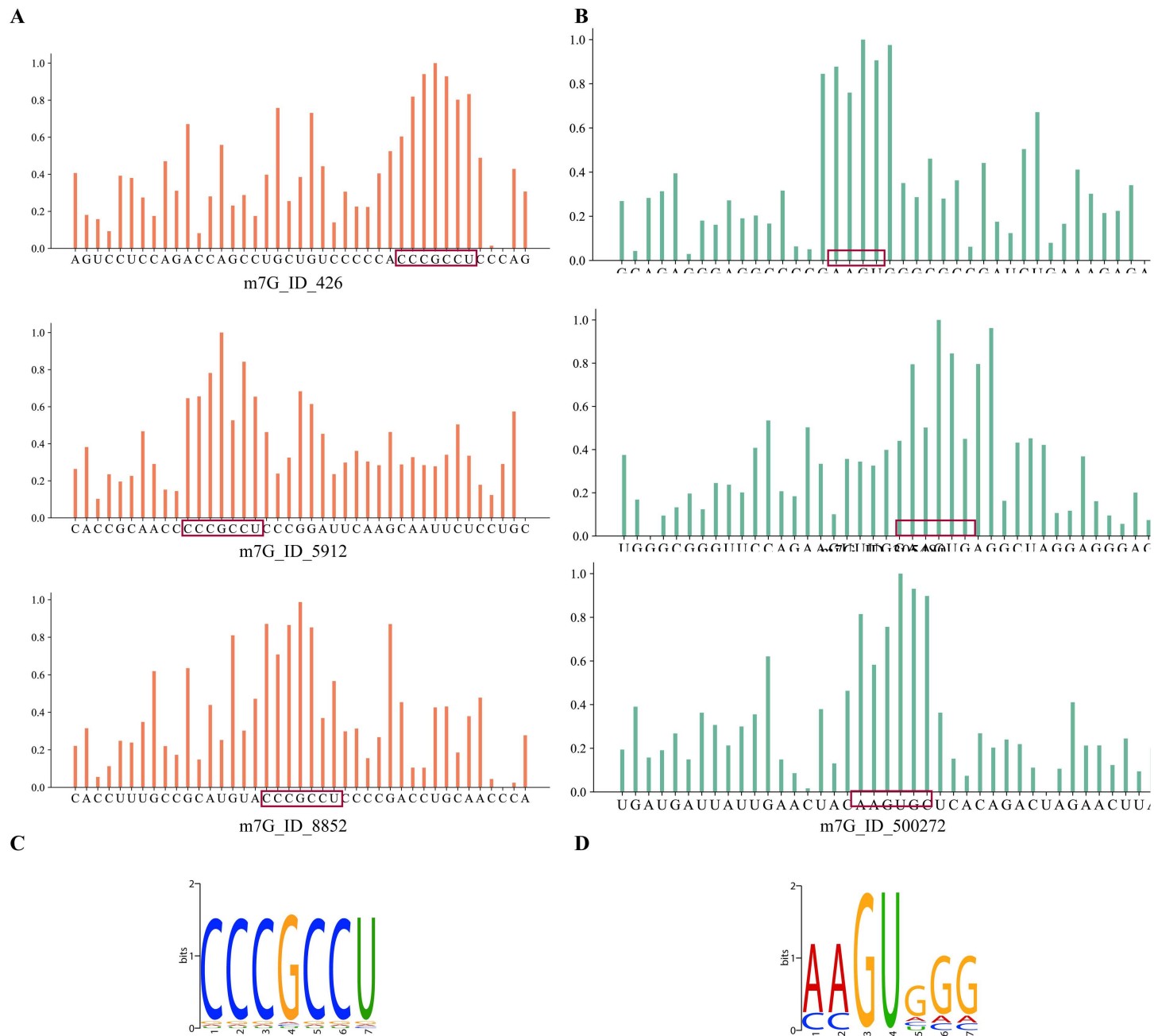

**Fig 7. Base importance of disease-related m⁷G host sequences and the conserved motif. A** Base importance of breast cancer-related m⁷G host sequences. **B** Base importance of Alzheimer's disease-related m⁷G host sequences. **C** Conserved motif of breast cancer-related m⁷G host sequences. **D** Conservative motif of Alzheimer's disease-related m⁷G host sequences.

Despite achieving promising results, PathoRM still has some limitations. Firstly, it solely solves the task of pathogenic RM sites prediction, ignoring the post-transcriptional regulatory procedures induced by pathogenic RM sites. Then the secondary structure of RNA may influence the pathology of RM sites. Incorporating the information about the secondary structure of RNA may enhance the predictive performance of pathogenic RM sites. Finally, regardless of m⁶A and m⁷G

modifications, PathoRM has the potential to be applied to a wide range of pathogenic RNA modifications, including $m^1A$, $m^5C$, ψ, and others. Furthermore, we envision PathoRM as a versatile tool for predicting various genetic modifications and interactions, such as pathogenic DNA methylation sites, miRNA-disease associations, and pathogenic non-coding RNAs. The flexibility of the PathoRM framework enables its adaptation to different types of biological data and research domains, making it a promising approach for studying diverse epigenetic and transcriptomic modifications.

## Supporting information

**S1 Text.** Optimization Procedure of Naïve Multi-View Learning; Message Passing and Updating Rules of Graph Neural Networks (GNNs); Metrics; PathoRM Outperforms the Traditional Machine Learning Models; Investigating the Optimal Sequence Length for Motif Analysis; Parameterization Analysis of PathoRM.
(DOCX)

**S1 Fig.** Comparison results of PathoRM and traditional machine learning models.
(TIF)

**S2 Fig.** Model performance with different site-centered sequence lengths.
(TIF)

**S1 Table.** 10-fold Cross Validation results of PathoRM and its competing methods on the m7GDA dataset.
(XLSX)

**S2 Table.** Statistical summary of model hyperparameters setting and trainable parameters.
(XLSX)

## Author contributions

**Conceptualization:** Lin Zhang.

**Formal analysis:** Jiani Ma, Lin Zhang.

**Investigation:** Lin Zhang.

**Methodology:** Jiani Ma.

**Project administration:** Hui Liu.

**Software:** Jiani Ma.

**Supervision:** Lin Zhang.

**Validation:** Xianjun Ma.

**Visualization:** Xianjun Ma.

**Writing – original draft:** Jiani Ma.

**Writing – review & editing:** Hui Liu, Jiani Ma.

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
