## [Decision Letter · Decision Letter 0]

21 Jan 2025

PCOMPBIOL-D-24-01052

PathoRM: Computational Inference of Pathogenic RNA Methylation Sites by Incorporating Multi-view Features

PLOS Computational Biology

Dear Dr. Zhang,

Thank you for submitting your manuscript to PLOS Computational Biology. As with all papers reviewed by the journal, your manuscript was reviewed by members of the editorial board and by several independent reviewers. In light of the reviews (below this email), we would like to invite the resubmission of a significantly-revised version that takes into account the reviewers' comments.

As you will see, the reviewers (and I) found the work valuable and important. However, some valid concerns were raised regarding the curation of the training dataset and the usability of the tool. In particular, the reviewers found the biological relevance of the manuscript difficult to evaluate and would have liked a more comprehensive background and discussion sections covering biological aspects, dataset curation, and usage examples, to make the manuscript relevant for less technical readers. They also ask for a detailed discussion on the decision to focus on m6A and m7G modifications. In addition, reviewers 1 and 4 had difficulties using the GitHub repository and would like to have better documentation, usage examples, and installation guidelines. I think that addressing these concerns will make the paper much stronger and its potential impact much greater. The authors are also asked to edit the language of the manuscript for clarity.

Please do assure addressing these concerns as well as any other issue raised by the reviewers before making your resubmission. We cannot make any decision about publication until we have seen the revised manuscript and your response to the reviewers' comments. Your revised manuscript will also be sent to reviewers for further evaluation.

Please submit your revised manuscript within 60 days Mar 23 2025 11:59PM. If you will need more time than this to complete your revisions, please reply to this message or contact the journal office at ploscompbiol@plos.org. Please include the following items when submitting your revised manuscript:

We look forward to receiving your revised manuscript.

Kind regards,

Aya Narunsky, Ph.D.

Guest Editor

PLOS Computational Biology

Shihua Zhang

Section Editor

PLOS Computational Biology

**Journal Requirements:**

At this stage, the following Authors/Authors require contributions: Hui Liu, Jiani Ma, Xianjun Ma, and Lin Zhang. Please ensure that the full contributions of each author are acknowledged in the "Add/Edit/Remove Authors" section of our submission form.

4) Please amend your detailed Financial Disclosure statement. This is published with the article. It must therefore be completed in full sentences and contain the exact wording you wish to be published.

1) Please clarify all sources of financial support for your study. List the grants, grant numbers, and organizations that funded your study, including funding received from your institution. Please note that suppliers of material support, including research materials, should be recognized in the Acknowledgements section rather than in the Financial Disclosure

2) State the initials, alongside each funding source, of each author to receive each grant. For example: "This work was supported by the National Institutes of Health (####### to AM; ###### to CJ) and the National Science Foundation (###### to AM)."

3) State what role the funders took in the study. If the funders had no role in your study, please state: "The funders had no role in study design, data collection and analysis, decision to publish, or preparation of the manuscript."

5) Your current Financial Disclosure states, "The author(s) received no specific funding for this work."

However, your funding information on the submission form indicates receiving funds. Please ensure that the funders and grant numbers match between the Financial Disclosure field and the Funding Information tab in your submission form. Note that the funders must be provided in the same order in both places as well.. 

Please indicate by return email the full and correct funding information for your study and confirm the order in which funding contributions should appear. Please be sure to indicate whether the funders played any role in the study design, data collection and analysis, decision to publish, or preparation of the manuscript.

**Reviewers' comments:**

Reviewer's Responses to Questions

Reviewer #1: Comments

In this manuscript, Liu et al. present a well-structured and organized computational workflow, PathoRM, designed for the identification of pathogenic RNA methylation sites. Compared with SOTA deep learning and machine learning models, PathoRM demonstrates significant potential in accurately inferring pathogenic RNA methylation sites and offers valuable insights for biological applications.

However, I have some concerns below.

1. The methods presented in this paper are highly complex, and the workflow is quite complicated. Therefore, the data calculations and variable symbols used in the algorithm are recommended to be consistently applied throughout the paper. Additionally, some symbols were not explained by the authors, which caused significant confusion for the readers. For example, XSD^, H(l) and LWCE, require further clarification.

2. I recommend that the authors provide additional clarification regarding the data used in this study. Exploring and experimentally validating the effects of varying data lengths would be beneficial. Specifically, for the RNA methylation data employed in the manuscript, further details on the length of the RNA sequences and the precise location of the RNA methylation sites are necessary. Moreover, the authors should investigate whether the length of the RM sequences impacts the predictive performance.

3. The authors should consider rephrasing some of the text in the abstract and introduction section. For example, the sentence "Here, motivated by this profound meaning, we developed a biologically informed deep learning model, named PathoRM, to help infer the associations among RNA methylation sites and diseases, providing convincing pathogenic RNA methylation sites and unraveling the enigma of pathology in the epi-transcriptomic layer” is quite long and can be split.

4. The authors released their code on GitHub but the repository lacks extensive documentation and instructions on how to run. I expect a repository with a well-explained readme and a sh script to install a virtual env where I can run the model.

5. Figure 8 needs to be reformatted. In some sub-figures, the red bounding boxes interfere with the xticks, while some are overlapped with the bars.

6. The authors are recommended to read through the paper carefully to better improve the presentation.

Reviewer #2: Comments

I have thoroughly reviewed the manuscript and found the study intriguing and of potential significance to the field. In this manuscript, Liu et al. proposed a computational workflow, PathoRM, for pathogenic RNA methylation sites identification. Equipped with LLMs, multi-view learning, GNNs and adversarial training, PathoRM achieves accurate the robust prediction performance across the metrics and dataset. Furthermore, PathoRM shows great potential in biological applications by capturing the conserved motifs that are related with pathogenic regions in the host sequence.

However, I have several suggestions to enhance the clarity and depth of the paper.

1.Dataset: The manuscript utilizes m7G and m6A as surrogates for RNA methylation in subsequent experiments. Could the authors elaborate on the rationale for selecting these two over others such as m1A, m5C, and PSI etc.?

2.10-fold CV: Two scenarios, balanced setting and imbalanced setting, were applied to validate the prediction performance of PathoRM. How do authors generate balanced dataset in the experiment?

3. Setpup: In Experiments, what kind of hardware facilities can support the operation of this article's experiment. Please explain in the experimental setup and environment preparation.

4 Figures: Figure 8 lacks consistency and alignment in the annotations of its subplots. Some of the bounding boxes overlap with the xticks, while others are positioned directly on the bars of the bar plot; Fig. 2 describes the “guilty-by-association”-based negative sampling scheme, which is good and informative. However, the figure’s caption must be written accurately and provided with a concise and useful explanation of the procedure.

5 Symbolic Notation In the “3 Implementation and Optimization“ section, what does Lwce represents? Please specify the meaning of it. In the “Adversarial Training Strategy for Model Optimization” section, the authors don’t specify the meaning of H(l)

6.Writing: The manuscript is generally well-structured, but there are a few long sentences that could be broken down for improved clarity and readability. For example, the sentence "Given the limitations in identifying pathogenic RNA methylation sites with in vitro experiments, there has been a need to develop computational workflow for the accurate inference of pathogenic RNA methylation to underpin biological investigations." is lengthy. Splitting it into multiple sentences may be useful and easy to read.

7.Equations: Some of the inline equations added to the manuscript need to be re-formatted.

Reviewer #3: This manuscript presents PathoRM, a computational deep learning model designed to predict pathogenic RNA methylation (RM) sites. The work incorporates advanced methods such as multi-view learning, large language models (LLMs), and graph neural networks (GNNs) and addresses a valuable topic with potential applications in disease treatment. However, several areas of the manuscript would benefit from significant revision to enhance clarity, biological grounding, and the justification of the model's design choices.

Strengths:

1. Innovative Model Design: The manuscript incorporates modern techniques, including BioBERT and iDNA-ABF, and introduces a unique multi-view learning approach that integrates heterogeneous features. This makes the manuscript useful for educational purposes.

2. Novel Use of Multi-view Learning: The construction of an affinity matrix by integrating semantic features from gene ontology (GO) terms and disease associations is novel, adding depth to the prediction framework.

3. Motif Validation: The validation of de novo predictions against known RNA motifs adds an experimental layer to the study, though this validation could be strengthened by testing motifs for additional diseases.

Areas for Improvement:

1. Biological Relevance and Background:

o Practical Applications of PathoRM: While PathoRM shows promise, it would be helpful to include real-world applications or case studies, showing its utility in disease contexts. This would make the work more relevant to readers outside the computational field.

o RNA Methylation Background: A more thorough introduction to RNA methylation and its biological implications is recommended before discussing the computational methods. This would make the study accessible to a broader audience, including those unfamiliar with deep learning.

2. Justification of Model Components:

o Complexity of Model Architecture: The manuscript lacks clear justifications for the many model components, such as the adaptation of iDNA-ABF (originally for DNA methylation) and the use of heterogeneous GNN structures with adversarial training. Since DNA and RNA methylation are distinct processes, further explanation is needed on why the iDNA-ABF module applies to RNA methylation.

o Adversarial Training and Negative Sampling: Both the adversarial training approach and the attention mechanism for normalization need more detailed explanations or illustrations to clarify their role in the model. This would improve the clarity of the computational approach.

3. Data Selection and Imbalance:

o Focus on Specific RNA Modifications (m6A and m7G): It would be helpful to explain the decision to focus on m6A and m7G modifications, given the data imbalance between the two. Disparities in sequence length (m7G: ~40 bp; m6A: ~64 bp) may affect feature quality and could explain why m6A outperforms m7G in cross-validation experiments. A more detailed justification here would strengthen the manuscript’s data approach.

4. Interpretability and Scope:

o Biological Interpretability: The manuscript mentions attention-based interpretability, but further detail on how attention scores correlate with known biological motifs or functions would make this clearer. For instance, validation of enriched motifs associated with diseases like breast cancer or Alzheimer’s against literature or known pathways would enhance interpretability.

o Prediction Scope: The model's predictions are limited to known RM sites and diseases, preventing novel discoveries. The term “de novo prediction” might mislead readers, as it suggests new site or disease identification.

5. Presentation and Structure:

o Clarity of Presentation: Abbreviations, formula components, input/output structure, and certain translation issues detract from clarity. We recommend careful proofreading to improve readability.

o Results Section Organization: A more hierarchical and optimized structure in the Results section would improve flow, making findings easier to follow.

o Comparative Analysis: Comparisons with unrelated models (e.g., drug-target and miRNA-disease models) are less relevant, while Figure 6 could be enhanced to better illustrate performance differences.

6. Expanded Validation of Motifs:

o We recommend extending motif validation to more diseases and considering recent literature on RNA methylation patterns. Additionally, highlighting novel, high-confidence predictions for validation by other researchers would further emphasize the model’s utility.

In summary, PathoRM introduces a promising computational framework for pathogenic RNA methylation site identification, but significant revisions are needed. A streamlined and clarified model design, with reduced complexity and clearer justifications, would likely yield a more interpretable and impactful study.

Reviewer #4: Very interesting method, with exciting results.

I would like to encourage authors to put bibliography on first occurrence of any specialist term, expand the discussion, and add comparison diagram for the methods discussed. Please also subdivide section 3 into different sections for description, benchmarking, and discussion.

Otherwise we have a majority of the paper in a lengthy section 3.

Most sections are subsections of 3. Please structure the paper better, since method description, benchmarking and discussion are recommended to be separated.

Regarding section 3.4: Are compared methods take all the same inputs? It should be emphasised.

Figure 1: Diagram for the methods compared and enumerated in section 2.2 is insufficient. One should also add a dataflow diagram with blocks for the methods described in the diagram. Such dataflow diagrams for other methods in comparison should be also added.

Figure 2: It would be nicer to see bar-chart comparisons of each category.

It is not clear which statistic is depicted in Figure 3 (P-value? E-value? Z-score?)

Since adversarial training scheme is attributed most of the performance, it would be nice to expand discussion and provide a simple pseudocode description.

Paper is light on possible generalizations. Is it because the discussed tool is not generalizing anything?

I recommend adding a section 3.13 on new possible pathological motifs, if found, or discussion why no new motifs were found.

Dataset described in section 2.3 looks fragmentary. Discussion of the completeness of dataset coverage, and estimation of the gains from the expansion is required.

The Figures 5 and 6 is very interesting and well made. I recommend checking it for colour blind readers though.

https://accessibleweb.com/color-contrast-checker/

Bibliography should be expanded to properly reference some key terms, like:

0. Entire introduction lacks bibliography for several statements:

* ("...to extract...lncRNAs, and targets".

* "..., BioBERT... acquired contextual understanding."

* "naive multi-view algorithm"

* "In spite of promise of some of the approaches..." (please cite at least once per approach enumerated)

* "...adeptly handling diverse data structures like..."

* "...thereby unveiling cryptic patterns..."

1. Xavier initialization strategy in section 3.3.

2. "matrix-decomposition derived decoder" at the end of section 2.5.

3. "PathoRM tries three GNNs, including GNN, GraphSage, and GIN."

4. Section 2.4 "CNF feature matrix" - either bibliography or expanding abbreviation on first use is required. Same for "disease GO feature matrix". (Not everybody is necessarily familiar with Gene Ontology etc.)

5. Please expand abbreviations on first use for "DOID and GWAS identifiers" in section 2.3.

Please note that missing page and line numbers make the submission somewhat inconvenient to review and correct.

As for the code, best practices dictate adding package metadata (`requirements.txt` file with the versions of packages needed to install and tested) and CI build to the GitHub repository.

**Have the authors made all data and (if applicable) computational code underlying the findings in their manuscript fully available?**

Reviewer #1: Yes

Reviewer #2: Yes

Reviewer #3: Yes

Reviewer #4: Yes

PLOS authors have the option to publish the peer review history of their article (what does this mean? ). If published, this will include your full peer review and any attached files.

**Do you want your identity to be public for this peer review?** For information about this choice, including consent withdrawal, please see our Privacy Policy .

Reviewer #1: No

Reviewer #2: No

Reviewer #3: No

Reviewer #4: No

**Figure resubmission:**
---

## [Decision Letter · Decision Letter 1]

25 Apr 2025

PCOMPBIOL-D-24-01052R1

PathoRM: Computational Inference of Pathogenic RNA Methylation Sites by Incorporating Multi-view Features

PLOS Computational Biology

Dear Dr. Zhang,

Thank you for submitting your manuscript to PLOS Computational Biology. After careful consideration, we feel that it has merit but does not fully meet PLOS Computational Biology's publication criteria as it currently stands. Therefore, we invite you to submit a revised version of the manuscript that addresses the points raised during the review process.

The reviewers appreciated your careful reply to their original review and agreed that in its current form, the manuscript is clear and presents important advances. However, some of their previous comments still did not get enough attention. Specifically, they suggested emphasizing the biological relevance of PathoRM, either by demonstrating its usage to study specific disease mechanisms or by revising the background and discussion sections to more comprehensively address how the tool can be used in the study of such examples. This could make the publication relevant to a larger community interested in computational biology.

Please do ensure addressing these concerns as well as any other issue raised by the reviewers before making your resubmission. We cannot make a final decision about publication until we have seen the revised manuscript and your response to the reviewers' comments. Your revised manuscript will also be sent to reviewers for further evaluation.

Please submit your revised manuscript within 60 days Jun 25 2025 11:59PM. If you will need more time than this to complete your revisions, please reply to this message or contact the journal office at ploscompbiol@plos.org. Please include the following items when submitting your revised manuscript:

We look forward to receiving your revised manuscript.

Kind regards,

Aya Narunsky, Ph.D.

Guest Editor

PLOS Computational Biology

Shihua Zhang

Section Editor

PLOS Computational Biology

**Journal Requirements:**

Please amend your detailed Financial Disclosure statement. This is published with the article. It must therefore be completed in full sentences and contain the exact wording you wish to be published. Please ensure that the funders and grant numbers match between the Financial Disclosure field and the Funding Information tab in your submission form. Note that the funders must be provided in the same order in both places as well.

**Reviewers' comments:**

Reviewer's Responses to Questions

**Comments to the Authors:**

Reviewer #1: All my comments have been properly addressed. I am happy to recommend the manuscript to be accepted at PCB.

Reviewer #2: Authors have addressed my early concerns

Reviewer #3: The authors have responded thoroughly to most reviewer comments, particularly around technical detail, code clarity, and model architecture. They added new analyses and reorganized the manuscript for improved readability. However, one key concern remains: the biological relevance and real-world utility of the model.

As stated in the journal’s scope, PLOS Computational Biology values research that not only develops novel computational methods but also provides new biological insights or shows potential for discovery. This manuscript still lacks a concrete demonstration of how the model advances biological understanding, even in a preliminary or illustrative way. The rebuttal defers this to future work.

In my view, this limits the impact of the paper within the journal’s mission. If the paper is to be accepted as a Methods or Research article, I believe a stronger emphasis on biological interpretability or a basic case study linking predicted methylation sites to a disease context would be needed.

I would support publication if the editorial team feels that the methodological novelty alone meets threshold, but from my perspective, it still falls short of delivering on the biological significance that the journal prioritizes.

Reviewer #4: I really enjoyed reading this article, but I found an important omissions that prevent understanding, verification, and reproduction of the research.

In particular, detailing number and width of neural network layers for each component of the PathoRM should always be reported. Standards for new AI papers require that the structure of the network is shown as a graph with each component having these numbers reported. Both dimensions, and activation functions should be reported.

For a great example of such report see figures 2 and 3 of "Going deeper with convolutions" of Szegedy et al.

Interesting tool for making publication quality neural network structure graphs is PlotNeuralNet of Haris Iqbal (https://github.com/HarisIqbal88/PlotNeuralNet). (The only clear indication of the number of layers was found by the reviewer in line 245, and only for the GNN part. While dimensions of BioBERT may be found in another publication, leaving them out makes the PathoRM paper feel like a supplementary to BioBERT paper referenced as [31], instead of a self-contained research report.)

Total size of parameters (in megabytes) should also be reported.

While training procedure seems carefully references, the omission of clear and complete description of network structure would prevent reproduction of this research and falls short of PLoS requirements for the data included.

Additionally the following supplementary information should be added:

A. Comparison of PathoRM statistcs in Figure 2A with the sum of ideal hyperbole corrected with a regression slope to assure

confirm or reject almost perfect fit of the results.

B. Clearly state whether different subnetworks fine-tuned (BioBERT) or trained separately, or all together?

C. Total size of training data and validation inputs mentioned in section 2.1.

Minor issues that reviewer have also found:

References for the methods used should be placed consistently after the method abbreviations instead of occasional shift to the end of the sentence.

In particular in line 98: Abbreviations DeepDTA and DeepDCA should be disambiguated with a paper reference placed immediately after each abbreviation.

Line 139: Paragraph starts with doubly subordinate sentence using passive mode that is extremely difficult to follow.

Figure 2.1 should be attached as vector graphics.

Line 322: "AUC metric" -> "The AUC metric"

Line 323: "LOODOCV" -> "the LODOCV"

Line 435: "distribution of AUC scores" -> "distribution of AUC statistic distributions"

Figure 5 between lines 447-450 should have Y axis clearly indicated as "distribution density".

Line 449: Description of figure 5 should state "Distributions of AUC statistic for..." instead of generic "Comparison Performance of ..."

Line 569: "Convervative motif" -> "Conserved motif"

Line 576: "positive ad negative" -> "positive and negative"

**Have the authors made all data and (if applicable) computational code underlying the findings in their manuscript fully available?**

Reviewer #1: Yes

Reviewer #2: Yes

Reviewer #3: None

Reviewer #4: Yes

PLOS authors have the option to publish the peer review history of their article (what does this mean? ). If published, this will include your full peer review and any attached files.

**Do you want your identity to be public for this peer review?** For information about this choice, including consent withdrawal, please see our Privacy Policy .

Reviewer #1: No

Reviewer #2: No

Reviewer #3: No

Reviewer #4: **Yes: ** Michał J. Gajda

**Figure resubmission:**

**Reproducibility:**



---

## [Decision Letter · Decision Letter 2]

23 Oct 2025

Dear Dr. Zhang,

We are pleased to inform you that your manuscript 'PathoRM: Computational Inference of Pathogenic RNA Methylation Sites by Incorporating Multi-view Features' has been provisionally accepted for publication in PLOS Computational Biology.

Best regards,

Aya Narunsky, Ph.D.

Guest Editor

PLOS Computational Biology

Shihua Zhang

Section Editor

PLOS Computational Biology

Reviewer's Responses to Questions

**Comments to the Authors:**

Reviewer #1: All my comments have been properly addressed. Happy to recommend the manuscript to be accepted at PCB.

Reviewer #2: authors have addressed my early concerns

Reviewer #3: We thank you again for your careful attention to the feedback and your thoughtful revisions. We hope these additional points will help guide any further improvements or future developments of this promising work.

**Have the authors made all data and (if applicable) computational code underlying the findings in their manuscript fully available?**

Reviewer #1: Yes

Reviewer #2: None

Reviewer #3: Yes

PLOS authors have the option to publish the peer review history of their article (what does this mean? ). If published, this will include your full peer review and any attached files.

**Do you want your identity to be public for this peer review?** For information about this choice, including consent withdrawal, please see our Privacy Policy .

Reviewer #1: No

Reviewer #2: No

Reviewer #3: **Yes: ** Namshik Han

---

## [Editor Report · Acceptance letter]

PCOMPBIOL-D-24-01052R2

PathoRM: Computational Inference of Pathogenic RNA Methylation Sites by Incorporating Multi-view Features

Dear Dr Zhang,

I am pleased to inform you that your manuscript has been formally accepted for publication in PLOS Computational Biology. Your manuscript is now with our production department and you will be notified of the publication date in due course.

With kind regards,

Zsofia Freund
